# Narratives of Survivorship: A Study of Breast Cancer Pathographies and Their Place in Cancer Rehabilitation

**Åsa Mohlin** [1,2,*] **and Katarina Bernhardsson** [3,4]

1   Department of Clinical Sciences Lund, Division of Medical History, Lund University, BMC,
    221 84 Lund, Sweden
2   Healthcare Center Laröd, Travvägen 27, 252 86 Helsingborg, Sweden
3   Birgit Rausing Centre for Medical Humanities, Lund University, BMC, 221 84 Lund, Sweden;
    katarina.bernhardsson@litt.lu.se
4   Centre for Languages and Literature, Lund University, 221 00 Lund, Sweden
*   Correspondence: asa.mohlin@med.lu.se; Tel.: +46-42-406-08-50

**Abstract:** The focus on cancer rehabilitation has increased, but breast cancer patients still report unmet rehabilitation needs. Since many women today will live long beyond their diagnosis, there are multiple challenges for the healthcare system in supporting these women in their new life situation. A more individualized approach is seen as necessary to optimize the rehabilitation for survivors. Pathographies, i.e., autobiographical or biographical accounts of experiences of illness, expose us to personal accounts of the journey through illness and treatment, offering us details, emotions, phrasings, and imagery from an individual perspective. In this literary study, we have analyzed two contemporary Swedish-speaking pathographies about breast cancer. In our analysis, we have presented perspectives on survivorship, and the authors' ways of conveying their breast cancer experiences through narrative. The pathographies envision the prominent impact the breast cancer has on the authors' lives. Narratives of survivorship have the potential to complement the more general medical knowledge with their nuanced and multifaceted stories of breast cancer. Learning from this type of material may improve the understanding of the complexity of breast cancer survivorship issues. This may be a way to become more attuned to identifying individual needs and preferences of breast cancer patients.

**Keywords:** breast cancer; cancer survivorship; rehabilitation; pathographies; patient narratives; qualitative; literary analysis; narrative medicine; medical humanities

## 1. Introduction

Breast cancer is the most common type of cancer in women worldwide [1]. A breast cancer diagnosis and its treatments are associated with various psychological and/or physical consequences across the demanding trajectory of the disease [2]. Since many women are likely to live long beyond the diagnosis, there are multiple challenges for the health care system in supporting these women in their new life situation. The focus on cancer rehabilitation has increased over the past years, but breast cancer patients still report unmet rehabilitation needs, and these unmet needs are related to reduced quality of life [3]. More and more, the need for an increasingly individualized approach is seen as necessary to optimize the rehabilitation for breast cancer survivors, but individualization is challenging to achieve in clinical practice [2,3].

In this article, we suggest a broadening of the materials and methods brought in to discuss these questions. In addition to more traditional assessments, e.g., surveys and interviews, we propose analyses of first-person narrated accounts of breast cancer survivorship. By studying these kinds of narratives, pathographies, we can bring into the discussion material that is singular and subjective, as well as rich in nuances and situated

in a specific context, as a complement to medical science's usual domain of more general, universal categories.

Pathography as a genre can be defined as autobiographical or biographical accounts of personal experiences of illness. It is a subgenre of autobiography with a specific focus on illness, treatment, and sometimes death [4,5]. The first-person perspective is privileged, putting emphasis on 'a person's own experience, on the thoroughly human environment of everyday life' in a manner similar to the philosophical study of phenomenology [6]. Illness is often the reason for the narrative—'Illness calls for stories', as Arthur W. Frank famously pinpointed [7].

Anne Hunsaker Hawkins coined the modern definition of the genre, and the proliferation of illness narratives has notably increased in the past decades, which have been called 'the golden age of pathography' [4,8]. This increase is linked to a larger cultural interest in 'confessional' stories of the self, a shift in media and publishing culture, and the movement in health care towards patient participation and the forming of patient organizations [9]. Pathographies are narratives where the ill person is in control of her story, of what to include and how to frame it. They are often written in a realistic way, and as Ann Jurecic emphasizes in her study of illness narratives, they demand from the reader a willingness to listen and connect [10].

Illness narratives of this kind appear frequently on best-seller lists, but many works have a more limited readership [5]. In Sweden, the first books published in the modern pathography genre were published in the late 1960s [11], and in the US slightly earlier [4]. Accounts of cancer make up the largest subgroup in the genre [11]. Forms and styles vary, including diaries, retrospective accounts, or poetry. Blogs and social media are also conducive to pathographic writing [12]. It is not unusual for photography and art to be included, and illness narratives can also be found in many other media [13,14]. Cultural analyses of illness may also have a pathographic dimension, taking the writer's experience as a starting point for a broader analysis; this has often been the case for breast cancer [15–17].

This study is conducted within the field of narrative medicine, a part of the broader subject area medical humanities, where pathography is a genre of interest alongside fictional works and other artforms [18,19]. We hypothesize that this kind of study can contribute new aspects in understanding the life situation of breast cancer survivors and help us become more attuned to listening to patients' individual needs and preferences. With this analysis, we wish to highlight the possibilities of using pathographies in both research and in the training of health care professionals to promote a more individualized approach in breast cancer care and rehabilitation.

The aim of this qualitative literary study is to analyse two Swedish-speaking pathographies about breast cancer with a special focus on life situations beyond the diagnosis and the future as breast cancer survivors. By studying these narratives, we are able to reach individual perspectives on illness experiences in the context of breast cancer survivorship in Sweden—findings that are of interest both in their own right and to showcase a type of analysis that can contribute to more multifaceted discussions on breast cancer survivorship onwards.

## 2. Materials and Methods

This study is placed within a hermeneutic, interpretative paradigm and uses a thematic, literary analysis to study the pathographies. This means that we share several goals with qualitative studies: we strive for knowledge of human experience, thoughts, and expectations, and are specifically interested in bringing out meaning and nuances [20]. The literary method of thematic studies means that our close reading of the works is specifically directed towards themes, which can be defined as 'middle-range textual elements that may be selected and identified by a reader because they are neither unique to only one text nor shared by much of world literature'. [21] Studying themes thus connects works in meaningful ways [21]. In a thematic study, there are many possible ways to structure

the analysis, and here, we are specifically interested in the pathographies' handling of illness through the theme of life as a trajectory, illness as a disruption of that trajectory, and rehabilitation and recovery as a possible return to a normal life. Within this theme, we explore treatment and waiting, survivors and community, images of illness and health care, and the body and its relationship to identity. These findings are structured under the headings 'Disruption of a formerly normal life,' 'Survivors and community,' and 'Recovery and rehabilitation—returning to a normal life?' Finally, in the discussion, we outline ways to make the narratives, and the exploration of the narratives, useful in a medical context.

These pathographies are not chosen to be representative of a specific group, but to provide an opportunity for the in-depth study of individual, literary narratives, and throughout the study we give detailed examples from the works. We study pathographies from our native Sweden, and the context for the narratives is thus the cancer care of a Nordic welfare society, where breast cancer is the most common cancer for women and the survivor rate after ten years is more than 80 percent. In recent years, more women are diagnosed with breast cancer—this is seen in all age groups, with the highest incidence in the 60–69 age bracket—but also more are recovering, due to progress in diagnostics and treatments [22]. This makes the role of breast cancer survivors even more central.

The scope of this study is two book-length personal narratives published in the last ten years: Agneta Klingspor's 'Stängt pga hälsosjäl' ('Closed due to health reasons', 2010) and Yvonne Hirdman's 'Behandlingen. 205 dagar i kräftrike' ('The Treatment. 205 days in the kingdom of cancer', 2019) [23,24]. In the following, all references to the books will be given in parentheses continuously through the text. All translations from the Swedish originals are our own.

The criteria for including these pathographies are that (1) they depict the experience of breast cancer and contemporary Swedish cancer care, (2) they are written by women with a long life's experience behind them: Klingspor was 64 years old when she fell ill, close to the median age to be diagnosed with breast cancer in Sweden [22], and Hirdman was 74, (3) the explicit reason for writing the books is the diagnosis and illness, and (4) the authors are placing the breast cancer experience within a context of reflecting on their own lives, both of them having written autobiographically before—Agneta Klingspor (b. 1946) is a well-known novelist whose debut in 1977 was a diary from her youth, and she has since repeatedly used her own experiences in her novels, and Yvonne Hirdman (b. 1943) is a professor of history with a special interest in the welfare state and gender, and has written about her own family's history as well as an autobiographical account she gave the label 'ego history' [25–29].

The form of Klingspor's book is short passages following the progression of the illness but is also constantly interspersed with memories. The book begins with the statement 'She squeezed it in November. Or was it during the summer, on the island of Björkö?' (p. 7), which immediately gives us the sense of delay, of postponing the inevitable. However, 'now' she's 'here', in a waiting room, which forms the beginning of her cancer journey. Half of the book's pages contain photographs of her surroundings, everyday objects, and interiors from the hospital, as well as television images and a few self-portraits. Even though Klingspor writes an autobiographical account, she makes use of the author's liberty in creating a distance to herself and the illness, most notably by writing about herself in third person and constantly referring to 'the breast' as its own entity, separated from herself. Hirdman, instead, marks the beginning of the story as the start of treatment. Her account is in the form of a diary, every entry labelled with the date of writing, from January to August, covering every day of the 205 days of treatment. The first page starts with the words 'Tomorrow, the treatment starts', and the book ends with a chapter called 'Afterwards'. She comments extensively on the debates in present-day media and brings up both historical events and personal memories. Still, the diary's main subject, and what it revolves around, is breast cancer and the way the treatment of that cancer affects her body.

## 3. Results

### 3.1. Two Pathographies about Breast Cancer Experience

'Stängt pga hälsosjäl' ('Closed due to health reasons', 2010) and 'Behandlingen' ('The Treatment', 2019) are book-length accounts following the breast cancer trajectory, allowing the authors to pay close attention to the ailing body as well as to locate the present time in relation to their whole lives. In this analysis, we are specifically interested in the perspective of survivorship, life after the shock of diagnosis. The analysis is divided into three parts, 'Disruption of a formerly normal life', 'Survivors and community', and 'Recovery and rehabilitation—returning to a normal life?'.

### 3.2. Disruption of a Formerly Normal Life

A prominent structure in pathographies is seeing illness as disruption. The narratives tell stories of a life disrupted from what is viewed as a normal, expected life trajectory, first by the illness itself, and then by the treatment of it [30]. This disruption is seen already in the titles of the books: Hirdman is pinpointing the exact number of days she is stuck in 'the kingdom of cancer' and Klingspor is signaling that her regular life is on hiatus in the same way shops put up signs about being temporarily closed. Klingspor states this closure as being 'due to health reasons', but she is misspelling 'reasons' as its homophone 'soul' ('skäl'/'själ'), thus making this a matter of her soul and mind as much as her body.

The disruption of breast cancer is sudden and changes all their plans. The wider world, with all its engagements, must wait. Klingspor—who writes about herself in the third person, as 'she'—concludes: 'her name is called again, the small world calls and she grabs her bag and walks there. Now, it is the body. Her body. The only one' (p. 8). This acute sense of embodiment emphasizes how, when the body is ailing, it demands attention—the person is reminded of the embodiment that is always a condition of human life, but that in everyday life is easy to ignore. Hirdman comments how her body before cancer used to be 'a body to command, not the enemy–the damned horrible thing, damn all of it.' (p. 25).

Klingspor envisions her life and its disruption as a line: 'In an instant the straight line was bent into 'it is leaning towards cancer', and from one second to the next her life was threatened, and she was put in a state of low-risk for relapse for the rest of her life' (p. 86). The comment about 'leaning towards' cancer, which is the doctor's attempt to convey uncertainty before the diagnosis is clear, also highlights the experience of a change of direction, and Klingspor repeats the phrase several times. She stresses that it is not so much the fear of death, but the fear of a new and unexpected kind of death that throws her. Her former image of death, 'to get older and die, a straight line between birth and death, an escalation of time until the body said goodbye in a heart attack or just gave out' is suddenly curved into the cancer threat (p. 86). She removes from sight tangible reminders of death, like the animal skull she used to have on her bookshelf.

Hirdman notes the division in her life by talking about a 'pre-cancer-Yvonne' (p. 163). She notes that she still most often thinks of herself as 'just a guest' in the kingdom of cancer (p. 162), but she also thinks of the experience as a 'dress rehearsal' of death (p. 23). When she has just received her diagnosis, she writes: 'I am thinking: We will probably have to cancel the trip to Finland./Is that all I am thinking? It is all I am thinking' (p. 16). The existential disruption is too large to immediately grasp, and is dealt with in a highly concrete form, becoming the first of many cancellations. Hirdman's illness narrative is focused on symptoms, pain, and losses. She comes back to the 'epicenter' of it all, her body (p. 89): she does not know 'if I even want to concern myself with thinking outside of my mutilated breast where the scar stings and my arm with the PICC pricks and itches and my horribly blemished arms and I don't have any eyebrows anymore' (p. 191)—and then, inevitably, she does concern herself. At the same time as she focuses on the body, her book is interwoven with thoughts about the history of the 20th century and her family. Again and again, she comes back to her father, her mother, her father's siblings, as if needing to formulate her personal history, her connectedness, when faced with possible death.

The treatment is a significant part of both narratives, but even more prominent is the amount of waiting and anticipating what is to come. Waiting and worrying seem to become the main form of existence. When events are postponed or cancelled, the impact on the authors is profound—any small obstacle brings despair to the surface. Hirdman was given incorrect information, a nurse promising sixteen days of radiotherapy ahead of her instead of the actual twenty-five, and her disappointment and resignation is substantial (p. 260). When Klingspor at one point has waited for information for a long time and is suddenly told there will be no news, she reacts strongly: 'One cannot call a cancer patient the day before an important notice to say there will be no notice, one cannot be uncertain of which day she will get notice, one cannot simply call like that', and then feels tears filling up (p. 60). The future, she writes, has disappeared from view: 'The future is no longer a time, the future is not a week or a month. The future is the results, nothing else' (p. 28).

While Hirdman depicts the individual caregivers as above all human and caring, the cancer treatment is highly technical and alien, featuring 'robot monsters', 'radiation monsters', and 'tentacles' (pp. 262, 270). The patients' bodies she envisions as 'lumps of meat on a conveyor belt' (p. 262) and the chemotherapy as being 'refueled' like a car, but with poison (pp. 64, 194). 'It is not only banal, this 'cancering', it is automatized too, industrialized. I am a treatment unit,' she writes (p. 188). The body and the handling of her makes her ambivalent: on the one hand she says it is not her body anymore, it has been transferred to the hospital and she will not acknowledge it, but then she confesses it feels nice, sometimes, not having to take any responsibility and to just show up (pp. 140, 256). In her first meeting with the hospital, Klingspor envisions the cancer patient as someone who cannot speak: 'Is she in the wrong place? Already the fact that she speaks in the mammography room feels like an aberration. Breasts do not speak, they are examined, and go out to wait for the result' (p. 9). Creating their own voice and their own story means resisting that feeling of being silenced, of becoming meat, and instead claiming the strength of an individual perspective.

The helplessness and loss of mastery over their body is originally the fault of the illness, but soon the cause is equally the health care system itself. Through the thinking—and most of all through the narrating, the control of writing the story—the authors can be seen taking back some of the mastery the illness denies them.

### 3.3. Survivors and Community

The importance of other people, especially other women, is continuously emphasized in the narratives. In many ways, the narrators give the impression of being lonely with their thoughts, but at the same time they have a network of other people—cancer survivors helping them to come to terms with the illness. It is only with the help of a friend Klingspor even takes the lump in her breast seriously, after ignoring it for some time: 'the lump became real through Ylva's reaction, it was created the moment she said it aloud and exactly as threatening as it perhaps is' (pp. 13–14). She carries the words from her friend Fateme, who later died from breast cancer, in her pocket, rustling as she walks, so she can take them out and read them: 'If I managed, you will manage' (p. 50). Hirdman describes talking with her fellow patients in the waiting room—'reveling in others' stories, sharing my own', creating a community (p. 278). The comparison with others ranges from deep sorrow to banality. Early on, Hirdman writes about her friend who sent her the message 'You have to laugh'. She comments: 'She is allowed to write that, she who lost her sparkling living daughter, mother of two small children, to this disease. Dearest Astri. But laugh?' (p. 9), and the sentiment of laughter is something she returns to several times. A few paragraphs later, the other side of the coin is shown in a conversation with an acquaintance whose mother survived breast cancer: 'It went well. It is so banal. Everyone has had it or knows someone who has. Cancer. Who has been through it' (p. 9).

The 'banality' of the illness is also mirrored in the struggle with the idea of writing about her illness. When a friend says, 'Just as long as you're not writing some cancer diary', Hirdman replies 'No, for crying out loud' (p. 104). The idea of taking part in a common

genre, perhaps even a banal one, where the stories are too similar, is troubling—but still, she writes. An important part of her narrative is this desire to move beyond the conventional. This is a strong pattern in both pathographies, the wish to go beyond stereotypes, to write the individual life truthfully and in detail. Hirdman thinks about acquaintances who have had a much worse time with cancer, especially her friend Christina whose comment 'why not me? Why should I escape it?' follows her through the illness and makes it impossible, even a taboo, for her to ask that common question 'why me?', even when she desperately wants to (pp. 62, 231, 234, 305). Hirdman's struggle, between the singular (her own pain and illness) and the general (an illness 'everybody' seems to have had), is palpable.

Klingspor, too, depicts her struggle with two points of view she cannot align. On the one hand, cancer is 'a fantasy, worse than reality' (p. 27), a view inspired by Susan Sontag's famous essay [15]. On the other hand, most patients survive: '80 percent live. Cancer equals death, however much she repeats the 80 percent. She has to put the sentences next to each other: 80 percent live after ten years and cancer equals death' (p. 27). The irreconcilability of these two statements is paralleled in how astonished she is to talk to cancer survivors from a patient organization: 'Surprised, she listens. As if she thought the cancer had silenced them, as if they were beyond the grave or had been there–is it possible to speak, then? She has never spoken to someone who has had cancer and continued their life' (p. 26). However, she immediately contradicts herself: she had met her friend Eva, 'with no hair and with a scarf on her bald head'. However, somehow, that meeting had still not filtered through fully: 'At that time, she herself was well and did not ask anything, she was part of the healthy world. It did not concern her. Despite Eva sitting in front of her with the side effects of the chemotherapy fully visible' (pp. 26–27). In a brutally honest way, she dissects the limits of empathy, the difficulty to fully understand. This is an insight Hirdman echoes—how empathy seems to need experience to be fully realized. As her friend Malin, to whom the book is dedicated, is going through chemotherapy again, she writes: 'I am aching from empathy. I think about her all the time. I would not have done that if I had not gotten "this"–cancer cancer. Then, the feeling would have been written, abstract–not really a feeling at all. Now I can hold it in my hand–squeeze it like an apple' (p. 115).

From these quotations, the authors suggest that an in-depth understanding of what others go through is difficult to achieve. At the same time, the empathy of people around them suggests it is possible, at least to some extent. Finally, we as readers are trying to reach that in-depth understanding through their perspectives and their experiences with cancer, following their path through the illness.

### 3.4. Recovery and Rehabilitation—Returning to a Normal Life?

Coming close to the end of the treatment, Hirdman writes: 'this–brutal?–treatment of my breast cancer is over after the radiation at 15.10′ (p. 296). The countdown towards this day, this precise moment, has permeated the pathography. However, only a few hours later, she breaks down. Instead of champagne and celebration, she is hit by the 'accumulated' exhaustion, collected during the long and tiring therapy. Despite the procedures having concluded, she writes: 'I still think–but soon. Surely, it is over soon?' (p. 297). Earlier, she asked herself: 'When do things become normal? Afterwards? But by then, things will never again become normal. I have aged. I am old' (p. 75). This is an important point—that our perception of normality can change both through aging and illness—and this kind of change can make it difficult to settle in new life situations. The difficulty is that just as the phase of reorientation starts, contact with the health care personnel usually ceases, as the cancer treatment has ended. The women who need support are thus often not identified by the healthcare system, but rather left to seek help by themselves. In the chapter 'Afterwards', Hirdman writes 'One day later, I write the banal words "it will never again be as it was". I look at the words critically. This platitude! But during the night, when the words came to me, they were thick, heavy with meaning' (p. 306).

There is a sadness in this, in having to reconcile oneself with the loss of the old existence. This process needs time after treatment; Hirdman repeats in the last chapter, 'Give it a little more time' (p. 315). Perhaps this process could have been eased by additional support from the healthcare system. The only presence from the healthcare system in Hirdman's 'Afterwards' is a box of anti-depressants that she never takes. She remembers the understated words from a psychologist she met during treatment, 'Afterwards, she said, one can become a bit depressed' (p. 313). No rehabilitation efforts or contact with the healthcare system are described. The only cancer rehabilitation mentioned during the whole narration is the physical one—'Move. All cancer rehabilitation talks about the significance of movement' (p. 197) she states, and later, 'I have to move the body so the soul becomes a little happier. But it is moping' (p. 205). There seems to be a lack of knowledge about cancer rehabilitation—which is much more than just physical activity—in her surroundings. This is not unusual in society, and studies show that within the healthcare system itself, the knowledge of how to create and combine rehabilitation efforts from an individual perspective, in terms of who needs what and when, is limited [2].

For Klingspor, the rehabilitation—or what she terms 'the healing', focusing more on the existential aspects—primarily takes place in nature and in the armchair where she sits reading a famous author's diary, being allowed to rest in 'the malstroem' of someone else's story (p. 91). She sits there for two weeks, more or less non-stop, hoping for the book to never end until, finally, 'the last page is looking at her and the breast' (p. 91). The possibility of losing herself in something is crucial, and Hirdman is looking for this too. She comments that writing, as well as reading, allows her to escape herself for a while: to become another, to create distance, which is a great comfort (p. 17). This escape is in both cases described as a kind of healing: a much-needed distance from the ailing body.

One of the important aspects of the survivorship is the body image. The body and especially the breast have irredeemably changed after the way they have been handled during the treatment. For a breast cancer narrative, the treatment's length as well as the magnitude of the interventions mean the stories are often transformative rather than in some simple sense 'restitutive' [7]. In breast cancer narratives, the breast is of course also central in the 'afterwards', as it has significance for self-image, body image, identity, and sexuality. The two narratives discussed here are written by older women, and this may be the reason why the breast is only partly used as a focal point for identity and sexuality, and in Klingspor's case in an especially interesting way. For Klingspor, a way of dealing with the illness is to separate herself from the breast. More and more as the story goes on, she talks about herself as 'she and the breast', noting the things they do together—she and the breast receive treatment, they take the bike to the forest, they read. It is a powerful way of narrating the story, on the one hand showing how her focus on the ill body part is so strong that she needs it to be always present and visible in her sentences, on the other hand, her way of writing makes the breast seem like something separate from herself. In this way, the breast is envisioned almost like another person, or a pet—something Klingspor can care for and even feel tenderness towards.

Klingspor also writes an emotional history of her breast, just before the operation takes place. She particularly focuses on it as part of her sexual self and of the maturing from girl to woman, exploring common images of breasts as well as creating new ones. She describes the novelty of having breasts in puberty—wondering whether they will get in the way of doing things—and following on this the beauty, the pleasure, and finally, the transformation to a 'death breast' (p. 55). This short history seems to function as a kind of farewell to the untroubled body of her earlier life. Hirdman describes a kind of farewell in her husband's gesture, as she during the night 'made your hand cup the breast that will soon not exist anymore–at least not in this form' (p. 18); this is the only time she uses 'you' instead of 'he' in the narration, which emphasizes the intimacy of the gesture. Later, Hirdman notes that her breast is reduced and likens it to a wrinkled old apple (p. 23), but despite the centrality of the breast, she focuses more other body parts throughout the book: the lost hair and eyebrows, the tingling feet, the arm where the PICC catheter is inserted,

the fingertips—fearing the ache in them may stop her from writing—and, most of all, the stomach repeatedly giving her trouble. While she does worry about her looks, her main worries are about the functionality of her body.

One of the recurring themes in breast cancer stories is the hair and the loss of it. Many stories about cancer make use of images and clichés around hair loss; it is possible to trace the use of wigs in cancer stories as a device that can be both playful and at risk of reinforcing stock gender expectations [31]. Hirdman has a pragmatic and rather unsentimental view of both her hair—'what if I finally would colour it red? Before it falls off?' (p. 23)—and of wigs, emphasizing how they are scratchy and ill-fitting. She freely admits vanity, especially as she gives public talks a few times during the treatment. At the same time, she returns several times to the feeling of touching her bald head and, later on, the short stubby hair, and finding it a pleasing sensation. She calls it 'egg head', likening it to her husband's head, and she sparingly uses wigs, preferring the other kinds of cancer headdresses. On the last page, she puts her wig away at the top of a bathroom cabinet—a simple gesture of freedom (p. 316). In Klingspor's case, her hair is spared—in the end it turns out she does not need chemotherapy. When she receives the news, she has just 'washed it this morning with a farewell in her hands', a comment highlighting the weight of the loss she had envisioned (p. 70). She has previously stated that cancer patients are invisible until they start to lose their hair or have prominent scars (p. 32); she can thus keep on being invisible and will not have to deal with any 'intravenous intake of poison' (p. 70). As she leaves the hospital, relieved, she passes a bridge and feels that she 'loves everything, every single algae and common roach in the Årsta bay' (p. 71).

One of the points Hirdman makes is about information—how difficult it is to receive, and how easily it is given at the wrong time, for example too close to the diagnosis, when the patient is still in shock, or too much information being given at the same time. Studies show this is not an uncommon problem, and that this discrepancy can make the transition to life after treatment more difficult [32]. The problem could be interpreted either as a human reaction to too much information or as a side effect of the illness and treatment, but Hirdman blames herself. She feels she has behaved like an ostrich: 'I realize I have done it again! Ostriching. Not finding some things out. Taken one stage at a time and forgot the others' (p. 166). One of the things she has ignored is the information that depression can come after the treatment has ended; she was not ready to think about that in the middle of radiation. Hirdman writes that when she received the diagnosis, they 'went home and were exhausted by all the information that had been poured over us during these long hours' (p. 18). After the treatment, she wonders about the ache in the lymph in her arm, is that really supposed to be there? She remembers they had received a whole lecture about the lymphatic system by an enthusiastic therapist, but at the time it was 'completely wasted on us' (p. 293). The timing had been bad, but also now, after a new meeting with the lymph therapist, she is still lost, having gotten a load of information but no answers to her most pressing questions. The queries rush around in her head, and she 'debates right out into the emptiness' (p. 294).

Neither of the stories end in a happy stance; rather, the endings are in different shades of grey, and not entirely easy to decipher. Klingspor emphasizes her own strategies to heal from the aftermath of illness, resting and healing in her armchair as she reads; recovery seems in many ways to be an individual task, although she is in contact with another person through his writing. In the end, as 'she and the breast' are by the ocean, she is again able to think about death and dead family members, no longer needing to flee every trace of death. She can even read the obituaries in the newspapers (p. 93). Hirdman refuses the two possible outcomes after the illness, 'back to normal' or 'transformation', criticizing the latter for being too heroic, and embellishing suffering in a masochistic way. She neither finds catharsis, salvation, nor a reward (p. 315). She ends her book by again bringing up her dead parents and worries that she had failed to maintain their graves. This may sound like a bleak ending, but there is something in the energetic writing that does not leave the reader feeling that depression and numbness will win. The next-to-last line talks about the

shadows, the ancestors, who are calling her. However, the very last line reads 'I guess we will see, I say. We will see' (p. 315). The quiet, but still hard-set determination of that line leaves the reader with the feeling she is not giving up, even as she is not writing a standard hopeful ending of her illness narrative.

## 4. Discussion

In this thematic literary study, we have analysed two Swedish-speaking pathographies about breast cancer, Agneta Klingspor's 'Stängt pga hälsosjäl' ('Closed due to health reasons', 2010) and Yvonne Hirdman's 'Behandlingen. 205 dagar i kräftrike' ('The Treatment. 205 days in the kingdom of cancer', 2019), with a special focus on breast cancer survivorship. We propose that this kind of literary analysis can be a complementary way to reach understanding and formulations of breast cancer survivorship issues that are valuable for working with breast cancer patients—work that calls for a more individual approach since this group of patients is large and heterogeneous with diverse rehabilitation needs across the often long and demanding cancer trajectory [2]. The field of cancer rehabilitation has grown over the past years in Sweden, but we do not yet have any specific guidelines for rehabilitation of breast cancer survivors. Internationally, there are some guidelines addressing these issues [32], but even as the focus on rehabilitation has increased, breast cancer patients still report unmet rehabilitation needs [2]. We suggest that engaging with patients' narratives in literary analysis can help bridge the gap between medicine's general knowledge and the singular world of the patient, and thereby contribute more multifaceted aspects on breast cancer survivorship in both research and health care settings.

In our analysis, we have presented perspectives on survivorship through two contemporary Swedish pathographies about breast cancer, and the authors' ways of conveying their breast cancer experiences through narrative. The pathographies clearly envision the prominent impact the breast cancer diagnosis has on the authors' lives. The narratives tell stories of disrupted lives and changed body- and self-images, and the difficulties to adapting to and accepting new life situations after the diagnosis. The struggle between the singular (their own illness and losses) and the general (an illness 'everybody' seems to have had) is palpable. These narratives demonstrate the individuality of the breast cancer experience and the need to listen to every patient. However, cancer rehabilitation tends still to be based more on medical indicators rather than on the patient's individual needs and preferences in today's medical- and treatment-driven healthcare system.

*Using Pathographies and Literature in Health Care Settings*

Pathographies, i.e., autobiographical or biographical accounts of experiences of illness, expose us to personal accounts of the journey through illness and treatment, offering us details, emotions, phrasings, and imagery from an individual perspective.

How can this kind of reading and understanding be of use? The most obvious answer—basic but important—is that personal narratives give us new insight into how different perspectives, contexts, and previous experiences influence understanding and action, and how that can be captured through language and narration. The narrative as an important tool to create and understand identity and life situations is stressed in many disciplines [6,18,33–36]. Narrative is a broad concept, and here, we have specifically focused on written, published, literary accounts, by authors with longstanding experience in autobiographical narration. Through these kinds of narratives, as well as other art forms, others' perspectives can be experienced anew, which is rewarding both cognitively, emotionally, and aesthetically.

A second answer is that pathographies of this kind can be used for teaching and training narrative medicine and medical humanities. Around the world, work is conducted to include insights, methods, and materials from the humanities and arts in healthcare education and healthcare settings. Narrative medicine and medical humanities are included in the curricula of many medical schools, as well as in continuing professional education. These disciplines contribute by exploring 'contexts, experiences, and critical and conceptual

issues in medicine and health care' [19]. In Rita Charon's words, the effective practice of medicine 'requires narrative competence, that is, the ability to acknowledge, absorb, interpret, and act on the stories and plights of others' [37]. The use of narratives and other artforms offers a way of enhancing these competencies [38–40]. Arts in health, a related field of research and intervention, in a similar vein explores how artforms and creative initiatives can be used in interventions, with their focus on patient groups [41].

These types of personal, well-rounded stories can be read and discussed in several contexts, for example in medical education, in further education, and within inter-professional teams. They make room for discussions about individual and contingent contexts of an illness, about the thoughts, preconceptions, and personal history that color a patient's understanding of her own illness. They also offer stories reminding us that the time spent in a healthcare setting is important, but it is ultimately only a small portion of the patient's time.

Reading and jointly discussing narratives may stimulate curiosity, commitment, insights about the importance of perspective, and a 'narrative humility' with regard to the patient [42,43]. Especially when narratives are discussed in a group, a seminar's social and inter-personal sharing of views and thoughts open up an in-depth exploration of the texts as well as one's own reactions—contrasting the narrative to one's own experiences, studying the phrasings to bring out not only the writers', but also the group members' thinking more clearly. Added to reading and joint discussion, a third method may be added to this work: the writing session, where the exploration and interpretation of texts and experiences is continued through creative/reflective writing [38,44,45].

These personal narratives offer both closeness, by showing events from another person's perspective and through her choice of words, and distance, by offering this through the medium of text. It is important here that the narratives used are not explicitly written to be pedagogic, but rather written to include what the writers feel is important in their lives. The explanations, the metaphors, the digressions, the parallels—all serve to convey personal experiences. This allows us to read multi-faceted depictions of life and of the role of illness in that life. The stories are not easily summarized and categorized, they are full of life and emotion, of imagination and idiosyncrasies—just like patients are. The singularity and non-representativeness of these stories can be seen as a limitation of this kind of material, but it is in equal measure its strength, allowing for complexity and an understanding of illness as lived experience. The narratives cannot give exact knowledge about the general patient, but they can open up insights about the singularity of every patient, and through reading and discussions make us feel and remember that in a tangible way.

## 5. Conclusions

Here, we have presented voices of breast cancer survivors through two examples of Swedish pathographies and their methods of conveying their experiences through narrative. These narratives show the individuality of the breast cancer experience and the need to listen to every patient. Patient narratives of survivorship have the potential to complement the more general medical knowledge with their nuanced and multifaceted stories of breast cancer. This may be a way to become more attuned to identifying individual needs and preferences of breast cancer patients. Learning from this type of material may help caregivers and researchers to better understand the complexity of breast cancer survivorship issues. If breast cancer survivors can be provided with more individualized information, support, and rehabilitation, they may more effectively handle their life situations after the diagnosis and improve their quality of life throughout their new life trajectories.

**Author Contributions:** Conceptualization, Å.M. and K.B.; methodology, K.B.; formal analysis, Å.M. and K.B.; investigation, Å.M. and K.B.; resources, Å.M. and K.B.; writing—original draft preparation, Å.M.; writing—review and editing, Å.M. and K.B.; supervision, K.B. Both authors have read and agreed to the published version of the manuscript.

**Funding:** This study was supported by grants from the Birgit Rausing Donation and the Faculty of Medicine, Lund University.

**Institutional Review Board Statement:** Not applicable as ethical review and approval are not necessary for the study of works published by book publishers.

**Informed Consent Statement:** Not applicable as consent is not necessary for the study of works published by book publishers.

**Data Availability Statement:** Not applicable. Literary studies do not include raw data.

**Acknowledgments:** The authors acknowledge Lisa Rydén, professor of surgery, and Anders Palm, senior professor in literary studies, for their support.

**Conflicts of Interest:** The authors declare no conflict of interest.

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
