# Peer review of "Narratives of Survivorship: A Study of Breast Cancer Pathographies and Their Place in Cancer Rehabilitation"

_curroncol, doi:10.3390/curroncol28040249_

Round 1

Reviewer 1 Report

This manuscript provided an interesting approach (ie pathographies) in providing insights into breast cancer survivorship.  I really found this quite interesting as I have no come across this type of manuscript in my readings in this space. 

My general comments:

  • I generally commend the articles on very good grammar and writing.  Please note that there is a grammatical error in line 13 of the Abstract that needs to be corrected
  • The primary hypothesis of this article is that a narrative medicine approach can contribute new insights in understanding the experience of breast cancer and better understand their needs.  My concern as a more traditional physician is not having a clear null hypothesis to have a "yes" or "no" answer to the scientific question.  Of course I understand this is often not feasible with this kind of qualitative research.  Therefore, I will defer to the Editors of this journal of whether this kind of manuscript fits within the scope of this journal
  • I felt it would be helpful to label or categorize some of portions of the narratives with the medical terms that are frequently used in describing concerns of breast cancer survivors.  For example, on page 8 around lines 369 - 376:  what is ultimately being described are body image changes and it might be helpful to integrate this kind of terminology.  Similarly, in the following paragraph (starting on line 386, the common symptom of cancer-related cognitive dysfunction (ie "chemobrain") is essentially described and might be helpful to integrate this kind of terminology. 
  • On Page 9, in the discussion, there is a reference to not having any specific guidelines related to breast cancer survivorship.  I would like to direct the authors to an ASCO Special Article published in 2018:  Lyman G et al, Integrative Therapies During and After Breast Cancer Treatment:  ASCO Endorsement of the SIO Clinical Practice Guidelines in the Journal of Clinical Oncology.  While it does not exactly capture all the needs described by the women in this manuscript, it does provide some meaningful evidence-based guidelines on integrative therapies to support some of the symptoms describes in the narratives in this manuscript.
  • Another consideration for the discussion section:  is it worth discussing the possible value of writing workshops for breast cancer survivorship?  For example, see:  Thomas R et al, Writing toward well-being:  A qualitative study of community-based workshops with breast cancer survivors in Can Oncol Nurs J, 2017. 

Author Response

Reviewer 1:

This manuscript provided an interesting approach (ie pathographies) in providing insights into breast cancer survivorship.  I really found this quite interesting as I have no come across this type of manuscript in my readings in this space. 

Answer:

We thank the reviewer for this kind and encouraging comment.

My general comments:

I generally commend the articles on very good grammar and writing.  Please note that there is a grammatical error in line 13 of the Abstract that needs to be corrected

Answer:

Thank you for finding this error, a word was missing in the sentence: “Since many women today will live long beyond the diagnosis”. We have updated the document.

The primary hypothesis of this article is that a narrative medicine approach can contribute new insights in understanding the experience of breast cancer and better understand their needs.  My concern as a more traditional physician is not having a clear null hypothesis to have a "yes" or "no" answer to the scientific question.  Of course I understand this is often not feasible with this kind of qualitative research.  Therefore, I will defer to the Editors of this journal of whether this kind of manuscript fits within the scope of this journal

Answer:

We understand that this kind of study does not lend itself to the same study design as traditional medical studies, and are very appreciative that there can be room for this kind of qualitative, even literary, study in this journal.

I felt it would be helpful to label or categorize some of portions of the narratives with the medical terms that are frequently used in describing concerns of breast cancer survivors.  For example, on page 8 around lines 369 – 376:  what is ultimately being described are body image changes and it might be helpful to integrate this kind of terminology. 

Answer:

Thank you for pointing this out. We talk about body-image and self-image in two places on a more general note in the article (line 337, 448 in the new version), and find it an important point. In the paragraphs in question, our aim to let the authors’ language choices and perspectives be at the centre, and not too much impose medical terms formulations on the quotes.

Similarly, in the following paragraph (starting on line 386, the common symptom of cancer-related cognitive dysfunction (ie "chemobrain") is essentially described and might be helpful to integrate this kind of terminology. 

Answer:

This is a very good point. In this instance, we are not only after the cognitive dysfunction but also the more general problem of giving information, and we have now rephrased to make this clearer (line 390-396 in the new version):

One of the points Hirdman makes is about information – how difficult it is to receive, and how easily it is given at the wrong time, for example too close to the diagnosis, when the patient is still in shock, or too much information being given at the same time. Studies show this is not an uncommon problem, and that this discrepancy can make the transition to life after treatment more difficult [32]. The problem could be interpreted either as a human reaction to too much information or as a side-effect of the illness and treatment, but Hirdman blames herself.

On Page 9, in the discussion, there is a reference to not having any specific guidelines related to breast cancer survivorship.  I would like to direct the authors to an ASCO Special Article published in 2018:  Lyman G et al, Integrative Therapies During and After Breast Cancer Treatment:  ASCO Endorsement of the SIO Clinical Practice Guidelines in the Journal of Clinical Oncology.  While it does not exactly capture all the needs described by the women in this manuscript, it does provide some meaningful evidence-based guidelines on integrative therapies to support some of the symptoms describes in the narratives in this manuscript.

Answer:

We have clarified in the text that we first were referencing the Swedish situation, and then after this we also bring it to the broader international context (line 438-439):

The field of cancer rehabilitation has grown over the past years in Sweden, but we do not yet have any specific guidelines for rehabilitation of breast cancer survivors. Internationally, there are some guidelines addressing these issues [32], but even as the focus on rehabilitation has increased, breast cancer patients still report unmet rehabilitation needs [2].

We lifted into this discussion reference 32, which belongs in this discussion and shows an international example of breast cancer guidelines. The reviewer’s suggested reference is very good, but we deem that the information is already covered in one of the other references, 2, which is a systematic review of what is requested.

Another consideration for the discussion section:  is it worth discussing the possible value of writing workshops for breast cancer survivorship?  For example, see:  Thomas R et al, Writing toward well-being:  A qualitative study of community-based workshops with breast cancer survivors in Can Oncol Nurs J, 2017. 

Answer:
This is a very good comment, this kind of work is definitely interesting and promising, and we would like to discuss this kind of work more in the future. Here, we mention it in only one sentence, since it is not the focus of the article. We have now added in three words to highlight it even more on lines 480 – 481: “Arts in health, a related field of research and intervention, in a similar vein explores how artforms and creative initiatives can be used in interventions, with their focus on patient groups [41].”

The reference 41 is a book that provides an overview of arts in health including several disciplines and artforms, which covers the topic also of the suggested article. 

Reviewer 2 Report

I enjoyed reading this paper. Part of the value of pathographies is their compelling voice from writers with the gift of the insight and/or the ability to convey it with the written word. I think that they also suggest critiques and worthwhile improvements to the support and care of cancer patients and survivors but that is another kind of question.

1. What is the main question addressed by the research?

An analysis of survivorship themes in 2 pathographies, a form of literature concerning illness narratives that has wide public reach.

2. Do you consider the topic original or relevant in the field, and if so, why?

Yes. I have seen firsthand the value of pathographies for patients and family members seeking to better understand breast cancer and other experiences of illness. For the general public (and also for trainees in psychooncology), they offer a better basis for psychoeducation, empathy and support (both emotional and practical) than, sad to say, that provided by the psychosocial literature (whether qual or quant) in cancer, given its inaccessibility. The medical community could benefit from an awareness of what these well-read books are saying, and the possible critiques that may lie within concerning the state of care. 

3. What does it add to the subject area compared with other published material?

It's rare to see such a narrative analysis in an oncology journal and I think there's novelty in that. I consider this a form of qualitative study that bridges the case study and literary analysis. 

4. What specific improvements could the authors consider regarding the methodology?

I don't require further improvements here, keeping in mind the journal audience. However if I had to suggest some then: 1. the method could be described in greater detail. To what extent did the authors consider their analysis hermeneutic, for example. Why these extracted themes and not others?; and 2.
the three broad categories extracted seem to relate to the cancer trajectory (diagnosis/pre-treatment, treatment, post-treatment/survivorship), which seems a rather standard way to approach things. The more/most interesting aspects of the paper relate to their analysis and comparisons of the 2 authors' observations and experiences, which is more nuanced than suggested by the overarching headings. This disconnect is rather odd.  

5. Are the conclusions consistent with the evidence and arguments presented and do they address the main question posed?

Yes and yes. I don't require changes here given that the manuscript as it is reads quite well to me, but if some response is being required here for possible improvement, then I would suggest to the authors that they might seek to better tie in the results of their analysis to the scientific psychooncology literature. For example relating to the depersonalisation of the parts of one's body; or the narrowed psychological focus that patients often take on to get through treatment, only to have the emotional effects come later; or the lack of any clear ending that many patients feel once they've been cleared; etc. 

6. Are the references appropriate?

Seems fine to me.

7. Any possible comments that you would like to be addressed, or possible minor issue revised, despite your overall "accept" recommendation for this manuscript.

Spell check here and there (e.g. "whould"). 

I might also suggest to the authors as they move fwd with their work to explicitly analyse pathographies for what they say about the health care system and the treatment of patients and families as they move through it. Some of this may be impt for heath professionals seeking to improve care. Also it may shed light about cultural similarities/differences in illness experience if the authors took a comparative approach.

Author Response

Reviewer 2:

I enjoyed reading this paper. Part of the value of pathographies is their compelling voice from writers with the gift of the insight and/or the ability to convey it with the written word. I think that they also suggest critiques and worthwhile improvements to the support and care of cancer patients and survivors but that is another kind of question.

Answer:

We thank the reviewer for this very kind and encouraging comment.

NB: Points 1, 2, 3 and 6 were left without any query from the reviewer and we thus leave it without any reply.

  1. What is the main question addressed by the research?

An analysis of survivorship themes in 2 pathographies, a form of literature concerning illness narratives that has wide public reach.

  1. Do you consider the topic original or relevant in the field, and if so, why?

Yes. I have seen firsthand the value of pathographies for patients and family members seeking to better understand breast cancer and other experiences of illness. For the general public (and also for trainees in psychooncology), they offer a better basis for psychoeducation, empathy and support (both emotional and practical) than, sad to say, that provided by the psychosocial literature (whether qual or quant) in cancer, given its inaccessibility. The medical community could benefit from an awareness of what these well-read books are saying, and the possible critiques that may lie within concerning the state of care. 

  1. What does it add to the subject area compared with other published material?

It's rare to see such a narrative analysis in an oncology journal and I think there's novelty in that. I consider this a form of qualitative study that bridges the case study and literary analysis. 

  1. What specific improvements could the authors consider regarding the methodology?

I don't require further improvements here, keeping in mind the journal audience. However if I had to suggest some then:

Answer:

These comments, even as they are not requirements, are very good and we are happy to receive them. We have made adjustments in order to take them into account, and below we respond to the different comments.

  1. the method could be described in greater detail. To what extent did the authors consider their analysis hermeneutic, for example.

Answer:

This is a question of space, we chose to state the method enough to make it clear but without diving too deeply into it, out of consideration of the journal’s readership. We consider the analysis completely hermeneutic, and as this is a complex area of discussion, we felt it would go too far to write about it at length; instead we state that this is the paradigm we are working within. We have now included “interpretative” to make it even clearer (line 90): “This study is placed within a hermeneutic, interpretative paradigm”

Why these extracted themes and not others?;

Answer:

There is potentially a large amount of themes that could be discussed, and some of the aspects we analyze could be highlighted as themes of their own. In the end, it comes down to the choices of the analysts. We wanted to specifically address themes of relevance to the medical context. We have made some further comments to clarify it even more (lines 98-99):

In a thematic study, there are many possible ways to structure the analysis, and here, we are specifically interested in the pathographies’ handling of illness through the theme of life as a trajectory, illness as a disruption of that trajectory, and rehabilitation and recovery as a possible return to a normal life.

We also added in the structure of the coming text, to make it even more accessible (lines 103-105):

Within this theme, we explore treatment and waiting, survivors and community, images of illness and health care, and the body and its relationship to identity. These findings are structured under the headings “Disruption of a formerly normal life”, “Survivors and community”, and “Recovery and rehabilitation – returning to a normal life?”

and 2.
the three broad categories extracted seem to relate to the cancer trajectory (diagnosis/pre-treatment, treatment, post-treatment/survivorship), which seems a rather standard way to approach things. The more/most interesting aspects of the paper relate to their analysis and comparisons of the 2 authors' observations and experiences, which is more nuanced than suggested by the overarching headings. This disconnect is rather odd.  

Answer:

This is an excellent comment and that would be a good way of doing it too. We agree that the text include many important points that are not covered by the subtitles. We wanted to have a clear structure that would be easy to follow, especially as this is an unusual kind of study for the journal, and to then within this structure make room for the individual, interesting thoughts and images from the authors.

  1. Are the conclusions consistent with the evidence and arguments presented and do they address the main question posed?

Yes and yes. I don't require changes here given that the manuscript as it is reads quite well to me, but if some response is being required here for possible improvement, then I would suggest to the authors that they might seek to better tie in the results of their analysis to the scientific psychooncology literature. For example relating to the depersonalisation of the parts of one's body; or the narrowed psychological focus that patients often take on to get through treatment, only to have the emotional effects come later; or the lack of any clear ending that many patients feel once they've been cleared; etc.

Answer:

This is a great comment, and these parallels are interesting to make. We do not think there is space to include this here and that it goes slightly outside the scope of this article, but we would be happy to work more on this in the future. 

  1. Are the references appropriate?

Seems fine to me.

  1. Any possible comments that you would like to be addressed, or possible minor issue revised, despite your overall "accept" recommendation for this manuscript.

Spell check here and there (e.g. "whould").

Answer:

This is fixed!

I might also suggest to the authors as they move fwd with their work to explicitly analyse pathographies for what they say about the health care system and the treatment of patients and families as they move through it. Some of this may be impt for heath professionals seeking to improve care. Also it may shed light about cultural similarities/differences in illness experience if the authors took a comparative approach.

Answer:

We are grateful for this comment, and this is definitely something to work on.
